# Patients Benefit from Liver Transplantation for Hepatocellular Carcinoma beyond Milan Criteria without Harming the Health Care System

**DOI:** 10.3390/cancers14051136

**Published:** 2022-02-23

**Authors:** Jan-Paul Gundlach, Michael Linecker, Henrike Dobbermann, Felix Wadle, Thomas Becker, Felix Braun

**Affiliations:** 1Department of General, Visceral-, Thoracic-, Transplantation- and Pediatric Surgery, UKSH Campus Kiel, Arnold-Heller-Str. 3, 24105 Kiel, Germany; jan-paul.gundlach@uksh.de (J.-P.G.); michael.linecker@uksh.de (M.L.); felix.wadle@uksh.de (F.W.); thomas.becker@uksh.de (T.B.); 2Department of Internal Medicine-Hepatology, UKSH Campus Lübeck, Ratzeburger Allee 160, 23538 Lübeck, Germany; henrike.dobbermann@uksh.de

**Keywords:** HCC, hepatocellular carcinoma, liver transplantation, matchMELD, financial burden, organ shortage, Milan criteria, UCSF criteria, Eurotransplant, extended criteria organs

## Abstract

**Simple Summary:**

Liver transplantation (LT) is the only definitive treatment to cure hepatocellular carcinoma (HCC) in cirrhosis. Unfortunately, the shortage of donor livers limits access to the best available therapy. As a consequence, transplant candidates undergo selection for transparent waiting list acceptance and priority using the model for end-stage liver disease (MELD). Allocation of standard exception (SE) points for HCC inside the Milan-criteria balances the underrepresented urgency by labMELD in these patients who are exposed to tumor progression. Moreover, patients with HCC outside Milan can undergo LT without SE and might benefit from extended criteria donor (ECD)-grafts. We hypothesized that LT for Milan-out patients is associated with a more complicated postoperative course, reflecting higher costs. Interestingly, we found that LT for patients with Milan-in and Milan-out had comparable donor risk index, clinical outcome and cost-effectiveness. In conclusion, LT with ECD-grafts can have the maximum benefit for selected patients and bear limited financial risk.

**Abstract:**

Liver transplantation (LT) is the only definitive treatment to cure hepatocellular carcinoma (HCC) in cirrhosis. Waiting-list candidates are selected by the model for end-stage liver disease (MELD). However, many indications are not sufficiently represented by labMELD. For HCC, patients are selected by Milan-criteria: Milan-in qualifies for standard exception (SE) and better organ access on the waiting list; while Milan-out patients are restricted to labMELD and might benefit from extended criteria donor (ECD)-grafts. We analyzed a cohort of 102 patients (2011–2020). Patients with labMELD (no SE, Milan-out, *n* = 56) and matchMELD (SE-HCC, Milan-in, *n* = 46) were compared. The median overall survival was not significantly different (*p* = 0.759). No difference was found in time on the waiting list (*p* = 0.881), donor risk index (*p* = 0.697) or median costs (*p* = 0.204, EUR 43,500 (EUR 17,800–185,000) for labMELD and EUR 30,300 (EUR 17,200–395,900) for matchMELD). Costs were triggered by a cut-off labMELD of 12 points. Overall, the deficit increased by EUR 580 per labMELD point. Cost drivers were re-operation (*p* < 0.001), infection with multiresistant germs (*p* = 0.020), dialysis (*p* = 0.017), operation time (*p* = 0.012) and transfusions (*p* < 0.001). In conclusion, this study demonstrates that LT for HCC is successful and cost-effective in low labMELD patients independent of Milan-criteria. Therefore, ECD-grafts are favorized in Milan-out HCC patients with low labMELD.

## 1. Introduction

Liver transplantation (LT) is a standard procedure in the treatment of hepatocellular carcinoma (HCC). The costs of HCC care are rising due to its increasing incidence [1]. Although LT is known to have the highest impact on quality adjusted life years (QALYs) in small tumor stages, costs for LT are also higher compared to other curative treatment options [1,2]. Nevertheless, reports on higher tumor stages are missing. In parallel with organ shortage and increasing costs of transplantation programs [3], the limits of the transplant indication are being expanded without taking financial resources into account. This also applies to HCC.

Organs are allocated depending on an individual patient risk (sickest first principle) by the Model for End-stage Liver Disease (MELD) system [4,5], which was introduced in Germany in 2006 [6]. A bonus point-reporting system by the German Medical Association, comparable to the standard exception (SE) by the United Network for Organ Sharing (UNOS), exists for patients who experience an inadequate representation of the severity of the disease with the laboratory (lab) MELD. Patients with a SE-HCC, the so-called matchMELD, receive 22 points. This corresponds to a 3-month probability of dying in labMELD patients of 15%. The MELD score is then upgraded at 3-month intervals, which corresponds to an increase of additional 10% 3-month probability of dying in each case. MatchMELD criteria are fulfilled for patients within the Milan criteria (MC-in) due to their very good prognosis [7]: MC includes a single tumor lesion ≤ 5 cm in size or up to three lesions ≥ 1 and ≤3 cm [8]. However, only tumors ≥ 2 and ≤5 cm (UNOS T2) qualify for the matchMELD SE [9]. The diagnosis is usually made entirely on the basis of image morphology using the Liver Imaging Reporting and Data System (LI-RADS) criteria [10].

Beyond the Milan criteria, LT can be carried out within defined tumor stages (e.g., University California San Francisco (UCSF) score) with comparable oncological outcomes [11]. However, due to their mostly low labMELD score, these patients have an inferior urgency for organ allocation compared to matchMELD patients and compete with a variety of benign liver diseases. We postulate that HCC patients with a matchMELD receive an organ faster and at a superior quality than that of the labMELD collective. The aim of this study is to investigate whether the treatment of patients without bonus points has a negative impact on the clinical outcome and costs.

## 2. Materials and Methods

In the present work, a retrospective evaluation of a patient cohort with the ICD 10 diagnosis code C22.0 (HCC) was carried out. Patients were treated in a curative approach between the years 2011 and 2020 at our University Transplant Centre. Non-HCC patients, patients with mixed CCC/HCC and patients with extrahepatic tumor growth were excluded from the study. Transplant procedures have been described in detail earlier [12].

Basic data, pre- and postoperative tumor classification, clinical course and follow-up were correlated with donor data and insurance provider data. The influence of the donor organ quality was analyzed using data from the Eurotransplant (ET) register, the type of organ allocation (primary organ vs. rescue allocation) and the type of organ (full organ, split organ or living donor LT (LDLT)). Data for the ET-donor risk index (ET-DRI: age of the donor, cause of death (COD) hypoxia, cerebral infarction, cardiac death (DCD—not allowed in Germany) or others, origin regional or beyond, cold ischemia period, µGT and rescue offer were taken from the ET register. The score is calculated using the following function [13]:


*ET-DRI = exp {0.960 ((0.154 if 40 ≤ age < 50) + (0.274 If 50 ≤ age < 60) + (0.424 if 60 ≤ age < 70) + (0.501 if 70 ≤ age) + (0.079 *

*if COD = anoxia) + (0.145 × if COD = cerebrovascular accident) + (0.184 if COD = other) + (0.411 if DCD) + (0.422 if *

*partial/split) + (0.105 if regional share) + (0.244 if national share)) + (0.010 × (cold ischemia time − 8 h)) + 0.06 ((latest lab*

*µGt (U/L) − 50)/100) + (0.180 if rescue offer)}*


Costs were evaluated according to cost categories and accounts in accordance with the specifications of the German Institute for the Hospital Remuneration System (Institut für Entgeltkalkulation im Krankenhaus (InEK)) calculation manual (4th version) [14] and compared with the remuneration calculated by the InEK.

Data protection law was taken into account when evaluating this study. The study was approved by the local ethics committee (D 523/21). The statistical evaluation of this work was carried out with the program SPSS^®^ 25 (IBM^®^, Armonk, NY, USA). A significance level of *p* ≤ 0.05 was assumed to be significant. The graphical representation takes place with the GraphPad Prism software (version 9.1.2; GraphPad software, San Diego, CA, USA) or for the creation of limit value optimization curves with the program MedCalc (Statcon GmbH, Witzenhausen, Germany). The median and its range is given below. Patient survival was evaluated using a Kaplan–Meier analysis followed by a log-rank test. Descriptive data analysis (number and relative frequency in the total population in %), linear regression analysis and logistic regression analysis were carried out as further statistical methods. 

## 3. Results

### 3.1. Study Group Description

Between 2011 and 2020, liver transplants were performed in 112 patients at our University Transplant Centre as part of the diagnosis of HCC. Due to a lack of insurance provider data for patients who were still in inpatient treatment on 31 December 2020, three patients were excluded. Seven patients with an incidental tumor finding in the explanted organ were also excluded. A total of 102 patients were included in the statistical analysis of this work. Table 1 gives details of the patient characteristics. The HCC diagnosis was based on image morphology except for biopsy in 19 cases (18.6%) before ET listing to rule out a CCC. Liver cirrhosis is the most common morphological expression of liver-cell damage as the cause of the development of HCC (*n* = 98; 96.1%). Leading etiologies of cirrhosis in our cohort were alcoholic liver damage (*n* = 42; 41.2%) and nonalcoholic steatohepatis (NASH; *n* = 11; 10.8%). Four noncirrhosis patients had other liver cell damage such as fibrosis. A positive AFP value was documented in 47 cases (46.1%; >8 ng/ml). Concomitant symptoms are depicted in Table 1.

### 3.2. Eurotransplant Listing

The listing on the ET waiting list took place after recognition of the SE criteria for HCC in 46 Milan-in patients (45.1%) with a matchMELD (Table 2). In the collective, 11 patients were transplanted as UCSF-in (10.8%). The time on the waiting list did not differ significantly between both groups (*p* = 0.881). As expected, tumor size and number of tumor lesions differed significantly within the groups (*p* = 0.005 and *p* = 0.004, respectively).

Only 2 patients (2.0%) had dialysis before LT and 36 patients (35.3%) received abdominal surgery, of which 16 patients received a liver resection (15.7%). Locoregional therapy methods were performed in the majority of the patients, with transarterial chemoembolization (TACE) being the predominant procedure in 65 patients (63.7%) and together with selective internal radiation therapy (SIRT) in 4 (3.9%), percutaneous ethanol injection (PEI) in 2 (2.0%) and radiofrequency ablation in 3 (2.9%) patients, respectively. One patient (1.0%) received isolated PEI. In detail, in the matchMELD cohort, 39 patients (84.8%) received TACE (between 1 to 9 sessions, median 2). In sequence with TACE therapy, two patients additionally received PEI (4.3%) and two patients additionally received radiofrequency ablation (4.3%). In the labMELD cohort, 35 patients (62.5%) received TACE therapy (between 1 and 23 sessions, median 4); in 4 patients with additional SIRT (7.1%) and in 1 patient with radiofrequency ablation (1.8%). One patient received one PEI therapy without other locoregional treatment procedures (1.8%). Of note, the patient with 23 TACE sessions was on the waiting list for 4.3 years. No pharmacological HCC treatment was given before transplantation in our cohort.

### 3.3. Perioperative Results

While 16 patients were allocated a primary offer via ET, for 86 patients (84.3%) a rescue offer was accepted. Acute liver failure and HU listing did not occur. Four patients received an organ after LDLT (3.9%), two patients received a split organ and two further patients received organs after domino transplantation (each 2.0%). The median DRI was for the entire collective as well as for the labMELD and matchMELD collective 2.27 (1.14–4.10), 2.29 (1.40–4.10) and 2.17 (1.14–3.09) and the donor age was 62 (6–86), 63 (10–86) and 61 (6–86), respectively. In total, 19 matchMELD patients (33.9%) and 22 labMELD patients (47.8%) received DDLT with a donor age ≥ 65 years. In histopathological examinations, between 0 and 25 foci were found (median 2), the diameter of the largest lesion was 14.0 cm (median 2.5). No tumor could be detected in four patients after locoregional therapy. The distribution of tumor sizes between the labMELD and matchMELD cohorts also differed significantly (*p* = 0.011). The largest tumor in the labMELD cohort was 14.0 cm and in the matchMELD cohort 9.1 cm. The medians in the groups were 3.0 cm and 2.4 cm, respectively. The median waiting time for a donor organ from the time of ET listing was 109 days (1–1556 days). After a previous resection (*n* = 16), a transplant was carried out 501.5 days later (20–1367 days). The median surgery time was 269.5 (185–497) minutes. The cold ischemia time (CIT) was 565 (135–1025) minutes for the entire collective. Antiviral therapy with the hepatitis B immunoglobulin (HBIG) was carried out in 11 patients (10.1%) due to a replicative hepatitis B infection or for hepatitis B positive donor organ (*n* = 4). Subsequent hepatitis B prophylaxis was given to 22 patients (21.6%). CMV prophylaxis was applied to 38 patients (37.3%). The median postoperative length of stay was 16 days (1–203 days). High urgency retransplantation were performed in the labMELD and the matchMELD cohort in 2 and 3 cases (*p* = 0.405).

Importantly, we did not find significant differences in the DRI as well as the donor age between the labMELD and the matchMELD groups (*p* = 0.697 and *p* = 0.905).

### 3.4. Follow-Up

The median follow-up of the patients was 3.25 years (0–10.00 years). Of 34 deceased patients (33.3%, median 1.27 years after transplantation, range 0–7.58 years), 10 patients (9.2%) died during their inpatient stay (median 12 days after surgery, range 0–202 days). As expected, a survival analysis broken down into preoperative Milan and UCSF criteria showed no significant difference (*p* = 0.193, Figure 1a), although patients with UCSF-in tumors and patients outside Milan and UCSF seem to have a lower survival rate. The hypothesis was then checked whether there was a difference in survival depending on the MELD listing. The survival analysis (Figure 1b) of patients who were assigned an organ with a labMELD and patients with a matchMELD showed no statistically significant difference (*p* = 0.759). This result did not change when looking at the survival after diagnosis. Of note, the current follow-up of the patients with an organ age of 86 years were 3 and 10 years.

### 3.5. Financial Calculation

The cost bearer data for the transplant stay was evaluated next (Figure 2). The median costs for a liver transplant were EUR 37,332 (EUR 17,224–395,944) with a calculated revenue of EUR 41,897 (EUR 31,586–383,020). This results in a median revenue of EUR 7725 per case. In the labMELD and matchMELD collectives, costs were at EUR 43,513 (EUR 17,837–185,011) and EUR 30,324 (EUR 17,224–395,944), revenue was at EUR 44,724 (EUR 32,078–206,717) and EUR 39,908 (EUR 31,586–383,020) and deficit was at EUR 6433 (EUR −106,149–47,876) and EUR 10,182 (EUR −33,362–40,482). All results did not differ significantly. In the present HCC cohort, donor criteria such as donor age and DRI had no influence on the reimbursement of the inpatient case (*p* = 0.917 and *p* = 0.126) or patient survival (*p* = 0.835 and *p* = 0.726).

In order to evaluate a potential influence of bridging treatment on cost-effectiveness between labMELD and matchMELD groups, we analyzed costs, revenue and deficit of TACE treatment procedures, as this remarks the predominant locoregional bridging treatment. For the labMELD and matchMELD cohorts, costs were at a median of EUR 9074 (EUR 1780–53,801) and EUR 5325 (EUR 1658–28,210), remuneration at EUR 11,440 (EUR 2151–68,745) and EUR 6089 (EUR 1786–32,391) and deficit at EUR 2489 (EUR −2120–14,944) and 1223 (EUR −1113–5193), respectively. While the costs and remuneration differed significantly between the labMELD and matchMELD cohorts (*p* = 0.041 and *p* = 0.034), the deficit did not differ significantly (*p* = 0.112).

### 3.6. Predictability of a Deficit after Liver Transplantation

For preoperative prediction of a deficient hospital stay, we performed linear and logistic regression analysis. In the multivariate linear regression, maximum labMELD (*p* = 0.037) and the labMELD at the time of transplantation (*p* = 0.044) were highly significant. The deficit increased by EUR 576 per increase in the maximum labMELD or by 580 EUR per increase for the labMELD at the time of transplantation. A cut-off analysis for the labMELD score was carried out using a receiver operating characteristics (ROC) curve. Figure 3 shows the ROC curves for the maximum measured labMELD (a) and the labMELD at the time of the transplantation (b). The Youden index is a MELD score of >13 points for the maximum labMELD and >12 points for the labMELD at the time of transplantation. Both results are significant at *p* < 0.001.

In the multivariate logistic regression, a previous hepatitis C disease (*p* = 0.031) and a hepato-renal syndrome as a manifestation of the liver disease (*p* = 0.030) were found to be significantly associated with a deficit remuneration. However, in particular, the DRI, matchMELD and the waiting time on the ET waiting list did not show any influence on the costs and reimbursement of a LT. 

Despite numerous postoperative predictors in univariate analysis, the multivariate logistic regression showed only the need for a new operation (*p* < 0.001), the reinfection with multiresistant germs (*p* = 0.020) and dialysis (*p* = 0.017) to be significantly associated with a deficit. The multivariate linear regression showed the duration of the transplant operation (*p* = 0.012) as well as the need of erythrocyte concentrates (*p* < 0.001) and fresh frozen plasma concentrates (*p* = 0.025) significantly associated with a deficit. The financial deficit increased by EUR 86.50 per minute of surgery and by EUR 1300 and EUR 680 per erythrocyte or fresh frozen plasma concentrate. To check the quality of the parameter “operation time” as a criterion for assessing a deficit case, a ROC curve was created (Figure 4). The Youden index is an operation duration of >273 min (*p* < 0.001).

## 4. Discussion

Costs of HCC treatment are rising [1]. Under certain conditions, HCC can be treated curatively with a liver transplant. While LTs are known to have the highest impact on QALY, it marks also the most expensive treatment option [2]. So far, only HCC treatment of small tumor sizes is evaluated concerning its cost-effectiveness [2,15,16,17]. However, since patients with tumor stages larger than the Milan criteria, such as within the UCSF criteria, were demonstrated to have comparable oncological survival after transplantation [11], its burden on the health care system should be considered. We therefore intended to compare the effect of LTs for patients within and beyond Milan criteria on its cost-effectiveness. While Milan-in patients are entitled to a matchMELD, patients with a larger tumor do not receive any additional points, so that their urgency on the ET list is lower. Differences in outcome concerning this regard were discussed [11], but the financial aspect was not focused so far.

We were able to demonstrate a comparable outcome within our cohort. Although we presented, as expected, a significant difference in tumor size and number of lesions between the labMELD and the matchMELD cohorts (*p* = 0.005 and *p* = 0.004, respectively), overall survival after transplantation were comparable between both groups (*p* = 0.759). Of note, cohorts divided between Milan-in, UCSF-in and even tumors beyond UCSF did not show a significant difference in overall survival (*p* = 0.193). Moreover, we did not find differences in the waiting time for a donor liver: the time of waiting on the ET list did not differ (*p* = 0.881). Contrary to our expectations, financial differences did not vary significantly (*p* = 0.204 and *p* = 0.492). However, earnings for the hospital were slightly higher with matchMELD patients. Interestingly, the median financial benefit in TACE treatment, which is the most commonly used locoregional treatment modality, was positive in both groups. As expected, the costs for TACE treatment were significantly higher in the labMELD group (*p* = 0.041). However, the remuneration was even more intensified in the labMELD group (*p* = 0.034). In contrast to our actual expectations, the deficit did not differ significantly between the labMELD and matchMELD groups (*p* = 0.112) and neither group resulted in a loss of the care provider. In general, the liver transplant program in HCC can be performed cost-effectively. In contrast, unpublished results of our working group show costs of EUR 82,569 ± 81,820 with a deficit of EUR 281 per transplant case for the entire transplant collective (2011–2016; *n* = 179), regardless of the primary diagnosis. 

In order to understand which characteristics distinguish patients from cost-effective to cost-intensive cases, we performed single and multivariate regression analysis. The maximum labMELD (*p* = 0.037) and for the labMELD value at the time of transplantation (*p* = 0.044) were identified as possible parameters to predict loss-making cases: by means of a ROC analysis, threshold values of >13 points for the maximum labMELD as well as >12 points for the labMELD at the time of operation were identified (each *p* < 0.001). For every increased MELD point, the deficit increases by EUR 580. This is in agreement with the results of other studies [18,19,20,21]. In an American cohort between 2002 and 2005, a comparable increase in costs per MELD point of USD 580 was demonstrated [20].

In our study, the clinical condition of the patient before transplantation was only significantly associated with a deficit in compensation with regard to hepatitis C as an underlying disease (*p* = 0.031) and the presence of a hepato-renal syndrome (*p* = 0.030). Of note, donor criteria such as donor age and DRI did not have an influence on the reimbursement of the inpatient case (*p* = 0.917 and *p* = 0.126) or patient survival (*p* = 0.835 and *p* = 0.726). However, a corresponding prediction of high costs depending on the DRI score was demonstrated for heterogeneous patient collectives [22,23,24].

Postoperative complications, such as peritonitis, are other factors that can increase case costs [20]. Using multivariate logistic regression, significant correlations with a deficit case were found for the need of further operations (*p* < 0.001), infection with multiresistant germs (*p* = 0.020), dialysis (*p* = 0.017) and, in the multivariate linear regression, the duration of the transplant operation (*p* = 0.012) as well as the need of erythrocyte concentrates and fresh frozen plasma concentrates (*p* < 0.001 and *p* = 0.025). In ROC analysis, an operation duration of 273 min was identified as a critical threshold value, whereby the financial deficit per operation minute increased by EUR 86.50. The need of blood concentrates (erythrocytes *p* < 0.001 and fresh frozen plasma concentrates *p* = 0.025) increased the deficit by EUR 1300 and EUR 690. Other postoperative complications did not provide any indication of the development of a cost-ineffective case. 

In a comparison of liver resection and LT, the literature shows a longer patient survival in favor of transplantation, although the costs are significantly higher [15]. A positive cost–benefit assessment was shown for patients with survival after transplantation > 2 years [15]. This supports the need for a preoperative assessment of survival after LT on the basis of demographic, clinical and donor-specific data. So far, in different calculation models, prognoses of the financial outcome of a LT in comparison to resection were performed, although, they only consider small tumors in Child-Pugh A and B cirrhosis [16,17].

### Study Limitations

This study has several limitations. First of all, it is a unicentric, retrospectively collected data query in which the results can differ due to variables that are not considered in the regression analysis. The results should be compared with those of other transplant centers, although the data protection principles for this are lacking. Due to the small number of cases of *n* = 16 resections before subsequent transplantation, a comparison between resection and transplantation is not possible on the basis of the available data and requires a larger cohort. In addition, financial analyses comparing RFA, TACE, resection and transplantation, especially in the case of tumor manifestations outside the Milan criteria, should be tackled.

## 5. Conclusions

Patients with HCC outside Milan benefit from LT and show comparable survival to MC patients without financial burden for the caregiver. Our HCC transplant program is cost-effective for the inpatient stay as well as for the locoregional bridging treatment with TACE. It can be stated that there is no economic effect in terms of awarding bonus points for the HCC. Regardless of monetary pressure, patients outside the MC can undergo LT depending on the medical indication. The use of ECD organs is highly effective and not a risk for either the patient or the hospital. The survival of our cohort with a follow-up of 3.85 years is 67.7% and the hospital mortality is low, showing 9.2%, which is within the upper limit of 20% tolerated by the German IQTIG (Institute for quality assurance and transparency in health care). In addition, due to scarce resources and steadily increasing health expenditure, it seems appropriate to consider financial aspects in order to reveal wrong financially motivated triggers.

## Figures and Tables

**Figure 1 cancers-14-01136-f001:**
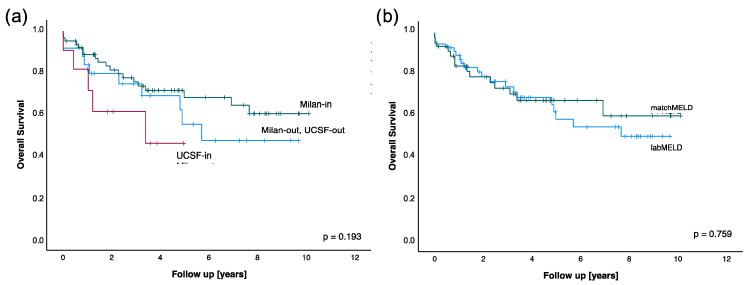
Kaplan–Meier analysis for overall survival in dependence of: (**a**) preoperative tumor criteria; and (**b**) labMELD and matchMELD listing.

**Figure 2 cancers-14-01136-f002:**
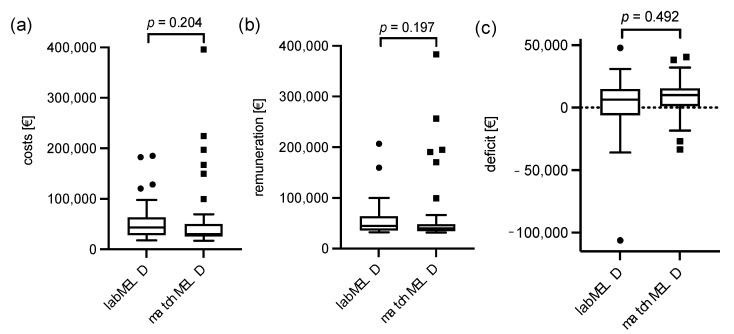
Cost analysis: Presented are (**a**) costs; (**b**) remuneration and corresponding (**c**) deficit for the labMELD and matchMELD cohorts in EUR (€), respectively. All differences are not significant.

**Figure 3 cancers-14-01136-f003:**
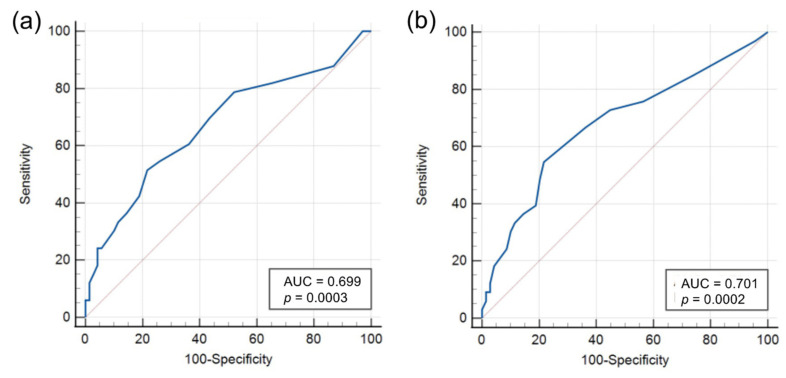
Critical value analysis for the labMELD. ROC-curve for (**a**): maximum measured labMELD and (**b**): the labMELD at the time of the transplantation for the detection of a cost-ineffective hospital stay. The Youden index is a MELD score of >13 points for the maximum labMELD and >12 points for the labMELD at the time of transplantation.

**Figure 4 cancers-14-01136-f004:**
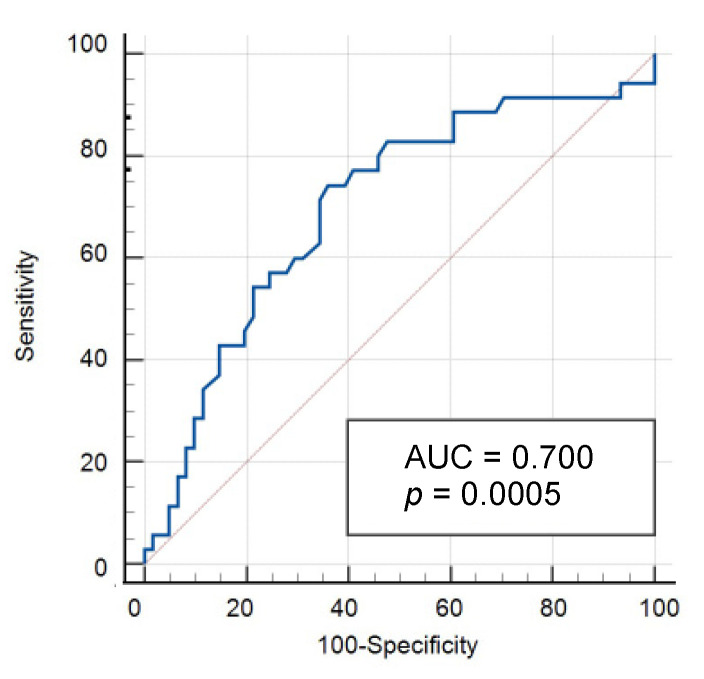
Critical value of the surgery time in ROC analysis.

**Table 1 cancers-14-01136-t001:** Patient characteristics.

Criteria	labMELD	matchMELD	*p*-Value
total	56 (54.9%)	46 (45.1%)	
Demographic data	*n* (%)	*n* (%)	
female	10 (17.9%)	13 (28.3%)	0.215
male	46 (82.1%)	33 (71.7%)	0.215
Biometric data	median (range)	median (range)	
age (years)	62.0 (30.4–74.3)	61.2 (44.5–73.4)	0.604
height (m)	1.78 (1.56–1.97)	1.75 (1.54–1.90)	0.067
weight (kg)	86.0 (48.0–154.0)	80.0 (51.0–140.0)	0.052
BMI (kg/m^2^)	26.78 (17.85–44.04)	26.13 (19.02–43.21)	0.180
Diagnosis	*n* (%)	*n* (%)	
cirrhosis	52 (92.9%)	46 (100%)	0.086
alcoholic cirrhosis	25 (44.6%)	17 (37.0%)	0.438
NASH	5 (8.9%)	6 (13.0%)	0.510
hepatitis	4 (7.1%)	4 (8.7%)	0.774
hepatitis A	3 (5.4%)	2 (4.3%)	0.816
hepatitis B/replicative hep. B	11 (19.6%)/1 (1.8%)	9 (19.6%)/4 (8.7%)	0.992
hepatitis C	11 (19.6%)	23 (50.0%)	0.001
hepatitis D	1 (1.8%)	1 (2.2%)	0.889
hepatitis E	1 (1.8%)	0 (0%)	0.367
Symptoms	*n* (%)	*n* (%)	
ascites	21 (37.5%)	12 (26.1%)	0.224
TIPS	7 (12.5%)	2 (4.3%)	0.152
hyponatremia (<130 mmol/L)	9 (16.1%)	8 (17.4%)	0.860
hepatic encephalopathy	11 (19.6%)	5 (10.9%)	0.229
hepato-renal syndrome	3 (5.4%)	4 (8.7%)	0.528
portal hypertension	25 (44.6%)	20 (43.5%)	0.907

Abbr.: TIPS—transjugular intrahepatic portosystemic shunt. *p*-values from univariate regression are displayed.

**Table 2 cancers-14-01136-t002:** Transplant criteria & MELD score.

Criteria	labMELD	matchMELD	*p*-Value
total	56 (54.9%)	46 (45.1%)	
days on the waiting list	109 (1–1556)	112 (2–1379)	0.881
size of largest tumor (cm)	4 (1–12)	3 (1–5)	0.004
number of tumor lesions	2 (1–5)	1 (0–3)	0.004
Milan-in	0	46 (45.1%)	
downstaged to Milan criteria	22 (21.6%)	0	
UCSF-in	11 (10.8%)	0	
UCSF-out	23 (22.6%)	0	
matchMELD score at LT		22 (22–34)	
labMELD score at LT	11 (6–36)	10 (6–34)	0.259
Donor risk index	2.29 (1.40–4.10)	2.17 (1.14–3.09)	0.697
donor age	63 (10–86)	61 (6–86)	0.905
reLT within 2 weeks	2 (3.6%)	5 (10.9%)/3 (6.5%)	0.405
reLT follow-up	0	2 (4.5)	0.117

Represented are number (*n*) and percentage (%) of the cohorts as well as median (range) for waiting time, diagnostic tumor size and MELD scores at liver transplantation (LT). For retransplantation, number of retransplantation within hospital stay of initial transplantation and number of retransplantation during follow-up are presented. All retransplantations within initial stay were high urge within 2 weeks. *p*-values from univariate regression are displayed.

## Data Availability

The clinical datasets supporting the conclusions of this study were derived from patient files (paper and electronic form). Therefore, restrictions to availability apply due to data protection regulation. Anonymized data are, however, available from the corresponding author on reasonable request and with permission of the University Hospital Schleswig–Holstein and the local review board.

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
