# Peer review of "Patients Benefit from Liver Transplantation for Hepatocellular Carcinoma beyond Milan Criteria without Harming the Health Care System"

_cancers, 2022, doi:10.3390/cancers14051136_

Round 1

Reviewer 1 Report

The article "Patients benefit from liver transplantation for hepatocellular carcinoma outside Milan without harming the health care system" is interesting and well-structured. Nevertheless;

  • the title should be reprhased; at least mention that you refer to "Milan criteria".
  • Importantly, information about patient pharmacological or radiological treatment is missing. These data would add significance to the manuscript.

Reviewer 2 Report

I read with much interest this original article by Gundlach and coll. entitled "Patients benefit from liver transplantation for hepatocellular carcinoma outside Milan without harming the health care system"

LT emerged as the best treatment option for HCC, and the development and widespread application of minimally invasive bridging and downstaging treatments contributed to further expansion of transplantability criteria.

Given this scenario, the current benchmarks for transplant eligibility shifted from a static dimensional assessment to a more dynamic evaluation of tumor biology and response to bridging treatments. 

The paper from Gundlach and colleagues is of particular interest as it focuses on the outcomes of LT form Milan-out HCC, taking into account the organ allocation and prioritization as well as the transplant costs.

The paper is well organized and well written

I have some requests to potentially improve the paper quality:

1) Provide the number of total labMELD and matchMELD in Table1

2) I noticed that the tumor size range of matchMELD cohort in Table 2 reaches 6, even if the matchMELD patients must fulfill milan criteria (and thus reach a max tumor size of 5cm); please explain

3) Please explain the p-value reported in line 187 (median 2.5; p = 0.011); which groups are your comparing?

4) Although the results from the study shows a comparable cost of LT for Milan-in and Milan-out HCC, I believe that the costs of the transplant procedure should be evaluated under an intention-to-treat perspective: for example, a Milan-out HCC undergoing repeated downstaging/bridging has a relevantly higher cost compared to a patient with a Milan-in undergoing LT ab-initio.
The Authors should comment on this behalf in the discussion section.

5) Please refer to the current literature concerning the evaluation of tumor-burden according to LI-RADS in the introduction section (10.1111/tri.13983)

Best regards.

Round 2

Reviewer 1 Report

I think that my concers have been properly addressed.

Reviewer 2 Report

the revised version of the manuscript is suitable for publication